# OpenReview forum: "Looking Inward: Language Models Can Learn About Themselves by Introspection"
_ICLR.cc/2025/Conference — ICLR 2025 Poster_

### Official Review · Reviewer_NSEX · 2024-10-29

**Soundness:** 3
**Presentation:** 4
**Contribution:** 3
**Rating:** 6
**Confidence:** 4

**Summary:**

The paper aims to show that Large Language Models have the capability to introspect, challenging the previous observation that these models merely imitate their training data.
In order to show this, the authors record the behavior of various models across a plethora of tasks as the ground truth, and fine-tune these models on the property-level behavior of a given target model. They then perform self-prediction, where the model is tasked to predict its own behavior, and cross-prediction, where the model is tasked to predict another model's behavior. The experiments show that the models are able to consistently outperform other, stronger models when predicting their own behavior.

In addition, further experiments show that the models calibrate towards their behavior distribution, even in cases where the fine-tuning data does not reflect this distribution. Finally, when fine-tuned on property-level tasks, the models can generalize to novel fine-tuning data, changing their behavior to match that of the fine-tuned model rather than the original one.

Overall, I believe this is a very interesting paper that sheds further light on LLM behavior. However, additional observations as well as analysis are required to cement the correctness of the experiments.

**Strengths:**

- The idea behind the possible introspection capabilities of Large Language Models is very interesting. However, I would be cautious regarding its framing as a non-data phenomenon as the introspective knowledge can arise from the training process and how the model was penalized with respect to next-token prediction during training for example.
- The paper is very well written, ample with examples and figures to bolster the claims, making it an enjoyable read.
- The experimental process is unbiased and comprehensive enough to showcase the behavior's truthfulness across models.
- The calibration analysis is especially interesting, and shows that significant behavioral distributions can be effectively learned by large language models.

**Weaknesses:**

- The definition of introspection (line 110) should be better defined. The fact that a better language model (I presume with respect to evaluation criteria) is not able to infer $F$ might not necessarily mean that $F$ can't be derived at all.
- When fine-tuning the $M_2$ model, does the training corpora exactly match that of $M_1$? If that is the case, isn't it possible that the prompt is eliciting a self-simulation even through fine-tuning in $M_2$? In other words, $M_2$ is generating answers based on its own behavior rather than those of $M_1$. I believe a further look into this would be valuable.
- Following the point above, a distinct advantage that $M_1$ has over $M_2$ is its knowledge of the introspection process, while $M_2$ is not provided with the knowledge that it is predicting another model's behavior during the training process. Do you think a significant change will be observed if we provide this knowledge to $M_2$ either through in-context knowledge or via fine-tuning with strings such as "Another model has predicted $X$ on $P$" where $P$ is the question prompt and $X$ is the generated answer?
- Although I agree that introspection is a valid explanation, I also disagree with your position in lines 357 to 363 where it is stated that it is not possible for introspection to partially arise from memorization. Can't it be that the fine-tuning process also allows the model to better align the question representation with that of the previously generated response?
- A further exploration of possible explanations would increase the work's merit. While the current explanation is appreciated, I think it would be good to expand upon this section.
- Consider condensing the figure captions to increase the available space in order to report more results in the paper itself rather than the appendix.

**Questions:**

- What do you think is the relation between the performance in the introspection task and the overall model performance? For instance, if a much stronger model is tested against a weak model (imagine GPT-4o versus Llama2 7B), can we expect the stronger model to infer the behaviors of the weaker model better than itself?
- How is the "mode" of output distribution is calculated as the baseline? Is this the mode with respect to the entire dataset under a single model?
- Why is it the case that a variant of COT can't improve this task on all levels? For example, prompting the model with something like "First generate the response that you would give to this question, and then choose the second character of your response" and then comparing to the ground truth. In other words, manually triggering the self-simulation.
- I suggest a better articulation for the phrases "property level" and "object level" prompts.

---

> ### Author Response · Authors · 2024-11-16
>
> We thank the reviewer for their detailed summary, which highlights our extensive experiments and interesting scientific questions.
>
> > a distinct advantage that M1 has over M2 is its knowledge of the introspection process, while M2 is not provided with the knowledge that it is predicting another model's behavior during the training process. Do you think a significant change will be observed if we provide this knowledge to M2 … via fine-tuning with strings such as "Another model has predicted X on P"
>
> The reviewer raises an excellent point about whether M2's performance might improve if explicitly told it is predicting another model's behavior. We conducted a new experiment with GPT-4o cross-predicting GPT-4, where we modified the training prompts to replace "you" with "another model" (the "Changed Pronoun" model).
> Results show no significant improvement in cross-prediction accuracy (34.9% → 35.7%), with the small difference likely attributable to variance between finetuning runs. This is still far below the self-prediction accuracy (48.6%).
> [This link shows the result. The Changed Pronoun model (purple) has similar accuracy to the cross prediction model (blue).](https://cdn.imgchest.com/files/l7lxcbod9w7.png)
> So the self-prediction advantage persists even when M2 is explicitly informed about its task, supporting our original conclusions.
> We thank the reviewer for this suggestion and have added these results to Figure 10.
>
>
> >I also disagree with your position in lines 357 to 363 where it is stated that it is not possible for introspection to partially arise from memorization.
>
> We evaluate on entirely unseen held-out tasks, ruling out direct memorization. Moreover, we test various *response properties*, where the model must output *properties* of responses rather than the responses themselves (e.g.  "second character", Fig 3). If memorization were the explanation, then M2 should predict M1's responses well after fine-tuning on M1's data. Instead, we find consistent self-prediction advantages (Fig 5). Most convincingly, in our behavioral change experiment, models update their predictions to match new behavior after fine-tuning, despite never seeing the new ground-truth to these properties (Fig 7).
>
> Are there other experiments you would suggest to further rule out memorization?
>
> >If a much stronger model is tested against a weak model (imagine GPT-4o versus Llama2 7B), can we expect the stronger model to infer the behaviors of the weaker model better than itself?
>
> GPT-3.5 results provide relevant evidence. We still observe a positive self-prediction advantage for GPT-3.5 (+0.8%, p=0.002) relative to GPT-4o. However, this advantage is very small.
> Since the models need to both reason about their behavior and extract particular properties of it, we hypothesize that self-prediction requires multi-hop reasoning (Figure 9), which weaker models are poor at (Yang et al., 2024b). Therefore, Llama2 7B may not show a self-prediction advantage because of weak reasoning abilities.
>
> > Why is it the case that a variant of COT can't improve this task on all levels?
>
> We agree that CoT should be explored in future work. But in our current setup, CoT would make the task substantially easier for models by decoupling self-simulation and reasoning about the result (as you point out) and therefore is less informative regarding measuring introspective capabilities. For that reason, we have focused on self-prediction in a single (or a few) forward passes.
>
> > How is the "mode" of output distribution is calculated as the baseline?
>
> The mode baseline is calculated per model, task, and response property. Added this clarification to Section A.4.2.
>
> >The definition of introspection (line 110) should be better defined. The fact that a better language model (I presume with respect to evaluation criteria) is not able to infer F might not necessarily mean that F can't be derived at all.
>
> Thank you for suggesting this. We acknowledge that for any existing model M2’s failure to infer F does not prove F is in principle non-derivable from data. We have amended our definition of introspection to clarify that a failure of any existing M2 to derive F is informative, but does not rule out that some stronger model could derive F.
> We also clarified that we test for introspection using a closely related criterion that can be empirically measured.
>
> >A further exploration of possible explanations would increase the work's merit.
>
> The reviewer and others have raised good questions regarding e.g. memorization, CoT and the Changed Pronoun experiments.
> We added discussions of these into the extended discussion section A.1.
>
> We appreciate your thoughtful feedback. In light of these clarifications as well as the Changed Pronoun experiment, would you consider increasing your score for our paper? If not, could you let us know any additional changes you would like to see in order for this work to be accepted?

---

> > ### Comment · Reviewer_NSEX · 2024-11-20
> >
> > Thank you for taking the time to respond fully to my points as well as the addition of new results plus discussion. Overall, I believe that the general idea behind this work has significant merit and can positively impact the community beyond that of model behavior interpretation, with possible use cases in model steerability as well as fairness/safety areas. As such, I am happy to increase my score to match these new findings. However, I believe that the following questions should be better answered in the future work:
> >
> > 1- Regarding fine-tuning -> While it seems like models have difficulty mimicking another model even through targeted fine-tuning (as done in the subsequent test), I would still like to know how resistant they will be to more aggressive information sharing between models. For example, through style transfer or if given access to the $M_1$ architecture or an answer database. All in all, further confirmation regarding the fact that this phenomenon is not due to pure data patterns would be significant.
> >
> > 2- Regarding memorization -> I agree with your point regarding the observation of the same behavior even in OOD data, but it can also be argued that the model has learned a data distribution under which a pattern is memorized, and as such, can simply generate tokens under the same distribution which matches its own generation pattern. More concretely, I think future work should address the hypothesis that the model is truly simulating itself under such cases.
> >
> > Overall, I think moving beyond the existence of such behavior and delving deeper into its interpretation is an important step to take in addition to this work.

---

> > > ### Author Response · Authors · 2024-11-22
> > >
> > > We are grateful that the reviewer finds that our general idea has significant merit and impact in interpretability, fairness and safety.
> > >
> > > >I would still like to know how resistant they will be to more aggressive information sharing between models.
> > >
> > > One experiment of interest is [(Figure 17)](https://cdn.imgchest.com/files/w7pjcpb3p97.png). We plot the data-scaling trend of cross-prediction. We find that the cross-prediction plateaus in accuracy after around 20,000 data samples. This suggests that the self-prediction advantage will hold even with more aggressive training on data samples.
> > >
> > > We thank the reviewer for suggesting other potential aggressive ways of information sharing for future work.
> > > As the discussion period is coming to an end, are there any further clarifications we could do to increase our acceptance beyond the marginal score?

---

### Official Review · Reviewer_fugu · 2024-10-31

**Soundness:** 3
**Presentation:** 3
**Contribution:** 3
**Rating:** 6
**Confidence:** 3

**Summary:**

This paper investigates the ability of LLMs to forms of self-prediction. They propose an evaluation setup for this and provide several experimental evaluations on several current LLMs.

**Strengths:**

The paper is generally well written and the experimental section showcase an in depth analysis.

**Weaknesses:**

I have concerns about the conclusions drawn in this work, particularly regarding the handling of “introspection.” Specifically, I am skeptical about the experimental setup and assumptions leading to the authors' conclusion that LLMs exhibit introspective abilities.

The results do not definitively support the notion of "introspection" as traditionally understood in cognitive science, where introspection entails self-awareness and access to one’s own mental states. Instead, the authors describe an experimental setup in which a model trained on its own outputs (M1) outperforms a second model (M2)—fine-tuned on M1’s outputs—in predicting M1’s behavior. This advantage is presented as evidence that the model has "privileged access" to its own behavior, which the authors relate to introspective ability.

However, this outcome can likely be attributed to the specifics of M1’s internal learned representations and operational dynamics rather than introspection. The observed advantage probably reflects M1’s better alignment with internal statistical correlations or patterns in its responses rather than a genuine introspective process akin to self-reflective thinking. In the absence of evidence against this alternative explanation (which I find lacking in the current setup), I consider it incorrect to conclude introspective abilities in any LLM. Therefore, while the results indicate an advantage for M1, interpreting this as introspection overstates the implications, especially as the task lacks the depth typically required to infer cognitive introspection.

In particular, the issues with more complex hypothetical questions (as the authors themselves remark on) indicate this interpretation. Specifically, the model’s limitations with complex questions provide evidence that its self-prediction advantage is bound to specific learned patterns rather than a flexible, introspective ability.

That said, I do recognize valuable insights in this work regarding LLM behavior that could be of interest to the research community. However, I find the conclusions drawn about introspection incorrect and potentially misleading. I recommend reframing the paper to remove conclusions involving “introspection” or “introspective” and replace them with more accurate terms. For example:

    - Self-Alignment: This term suggests the model’s responses align with patterns it has developed internally based on its training, without implying self-awareness.
    - Self-Predictive Consistency: This highlights that the model’s advantage arises from its capacity to predict properties of its responses based on embedded patterns, not metacognitive processes.
    - Behavioral Consistency: This straightforward term underscores the repeatability and stability in the model’s response patterns, which is what the tests measure.

Overall, I am concerned about labeling the behavior showcased here as “introspection,” a term with a well-defined, established literature and test designs from psychology. If the authors wish to call this a form of introspection different from that recognized in psychology, it is risky to attribute similar cognitive abilities based on a single type of evaluation. In light of these concerns, I recommend two changes: first, a reframing or rewording of the paper, which would be a crucial and achievable adjustment. Additionally, further experiments could be designed to challenge the intuition I’ve provided about potential underlying processes; however, I do not have specific suggestions at this time.

**Questions:**

N/A

---

> ### Author Response · Authors · 2024-11-19
>
> Thank you for your thoughtful comments. We are grateful you found our writing clear and our experimental analysis thorough.
>
> >a model trained on its own outputs (M1) outperforms a second model (M2)—fine-tuned on M1’s outputs—in predicting M1’s behavior.
> >However, this outcome can likely be attributed to the specifics of M1’s internal learned representations and operational dynamics rather than introspection. The observed advantage probably reflects M1’s better alignment with internal statistical correlations or patterns in its responses rather than a genuine introspective process
>
> This is an alternative hypothesis about our results. We respectfully disagree that this hypothesis is “likely”. We specifically designed our experiments to test this hypothesis.
>
> In all experiments, we evaluate models on unseen held-out tasks to prevent models from using correlations within each task to achieve high accuracy (line 214, S2.2). We also train and test models on *properties* of responses (rather than responses themselves). This means that models do not observe any actual responses during training [(Fig 3)](https://cdn.imgchest.com/files/myd5cd982g4.png).
>
> First, consider the Cross-Prediction experiment (mentioned above by the Reviewer). If model M1’s advantage was explained by statistical correlations, then we should observe negative results. Recall that in this experiment we train model M2 on M1's ground-truth behavior and test if M2 can predict M1's behavior (Section 3.2). If statistical correlations were sufficient, M2 should predict M1's responses well after fine-tuning on M1's data. Instead, we find clear self-prediction advantages across all pairs of models [(Fig 5)](https://cdn.imgchest.com/files/j7mmceqdxd7.png). We see the same self-prediction advantage on tests for behavioral calibration on held-out tasks [(Fig 6)](https://cdn.imgchest.com/files/l7lxcojka67.png).
> Finally, in our Behavioral Change experiment, models update their predictions to match their new behavior after an intervention, rather than to the distribution they were originally finetuned on [(Fig 7)](https://cdn.imgchest.com/files/l4nec6bx8v4.png).
>
>
> >Specifically, the model’s limitations with complex questions provide evidence that its self-prediction advantage is bound to specific learned patterns rather than a flexible, introspective ability.
>
> As noted above, models succeed in predicting properties of their behavior on six held-out tasks. They exhibit calibration on held-out tasks, despite being trained only on temperature=0 samples (Fig 6). They also succeed on the Behavioral Change experiment, which explicitly tests whether models can adapt their self-predictions in a flexible way (when the new behavior differs from specific learned patterns). We think this is substantial evidence for an ability that is not “bound to specific learned patterns”.
>
> We acknowledge that we see limitations with complex questions. This is an exploratory extension beyond our core experimental setup. We did not train models to tackle these more challenging tasks, so it is not especially surprising that they fail. These tasks involve new sources of complexity (e.g. much longer responses) than in the tasks we trained on. We suspect that any introspection ability that can be found in LLMs will have some limitations, but this does not undermine the claim that models have some introspective abilities. Likewise, LLMs show some ability to perform basic mathematical tasks via Chain-of-Thought, while currently having limited abilities in professional-level mathematics.
>
> >The results do not definitively support the notion of "introspection" as traditionally understood in cognitive science, where introspection entails self-awareness and access to one’s own mental states
>
> The term “introspection” covers a wide range of abilities in use in cognitive science. We do not assert or imply in the paper that LLMs have all of these abilities. We state in the abstract that we investigate “a form of introspection in which LLMs predict properties of their own behavior in hypothetical situations”. We then immediately describe our experimental setup. In the Introduction and Section 2, we try to convey exactly what abilities are tested in the paper, in order to clarify our claims about introspection. The Introduction (and Section 5) also highlight limitations in introspective abilities of LLMs.
>
> We agree with the Reviewer that it’s valuable to relate and compare our work to previous literature in related fields. The paper already related our use of “introspection” to use in analytic philosophy (Schwitzgebel 2024). We have now added a section (A.3.2.) that explains the connection in more detail. Moreover, as suggested by the Reviewer, we have added a section (A.3.1) that discusses use of the term in cognitive science and psychology.
>
> We thank the reviewer for their detailed review. Do you have additional suggestions to rule out statistical correlations as the reason for our results?

---

> > ### Author Response · Authors · 2024-11-22
> >
> > Dear Reviewer, as the discussion period is coming to a close soon, we wanted to follow up once more to check if we have addressed your concerns about statistical correlations?
> > We would be keen to use the remaining time to discuss improvements so that our paper could be better accepted.

---

> > > ### Comment · Reviewer_fugu · 2024-11-23
> > >
> > > Thank you for the response and the updated supplements. I think my point wasn’t so clear. I am not considering statistical correlations within tasks, but rather that M1 simply converges to similar states due to its parameter space. Finetuning M2 with M1’s responses won’t lead to M2 adapting such parameter spaces.
> > >
> > > One option that comes to my mind for additional experiments:
> > > - Incrememtally remove layers in M1 to observe in which manner the identified properties break. E.g. correlate the removal of certain model structures with the outcome.
> > > - Reinitialize M1 and perform the same protocol with several different seeds (this of course is almost impossible to perform with such large and commercial models), i.e. see how much the specific parameter space influences the results
> > >
> > > However, I am not urging the authors to perform new experiments, the paper already has a sufficient amount.
> > > So, all in all I do consider the results to be interesting and worth publishing. My point is mainly the phrasing and in my opinion overstatement of claims (i.e., introspection). For me the results indicate “consistent self-predictions” which is an important finding nonetheless. Rather than running new experiments I simply encourage the authors to change the claims and the title in such a way. The paper can still be discussed in the context of introspection, just the claim should be changed. In other words if the authors state “a form of introspection in which LLMs predict properties of their own behavior in hypothetical situations” then statements like “This provides indirect evidence for introspection” or “We provide evidence that LLMs can acquire knowledge about themselves through introspection” are overstatements. I had stated this and similar suggestions on phrasing in the last response, however the authors have not reacted to this. An alternative would be to name the specific form of introspection (which in my understanding is consistent self-prediction) and use that term throughout instead of introspection. Or even consider introspective capabilities as [2] does.
> > > A possible title change (just an example): “Looking Inward: Language Models provide Consistent Self-Predictions”
> > >
> > > [2] Long, R. (2023). Introspective capabilities in large language models. Journal of Consciousness Studies, 30(9-10), 143-153.

---

> ### Author Response · Authors · 2024-11-25
> **Changed introspection to introspective capabilities**
>
> Thank you for your thoughtful feedback.
> We appreciate your affirmation that our results are worthy of publication and your suggestions for more precise framing. We also agree that we should be careful to not overstate and be precise with our claims.
>
>
> We've revised our paper to be clearer that we investigate "introspective capabilities" rather than “introspection”, as you point out.
> The abstract should be more clear that we test primarily for the “privileged access” introspective capability. We also have added a statement “However, while we successfully show introspective capabilities in simple tasks, we are unsuccessful on more complex tasks or those requiring out-of-distribution generalization.”
> [For easy viewing, here is an image of the abstract. Red words show the changes.](https://cdn.imgchest.com/files/6yxkc9olr87.png)
>
>
> In our Overview of Methods (Section 2) we  added text to clarify that in psychology and philosophy, introspection refers to a broader range of capabilities e.g. self-awareness (which you’ve pointed out).
>
> We point out specifically there that we do not claim to show evidence for e.g. emotions and self-awareness.
>
> Method section adjustment:
> > We investigate an **introspective capability** in language models -- privileged access. Introspection often refers to a broader range of capabilities such as emotions and self-awareness which we do not study. Appendix A.3.1 and Appendix A.3.2 discusses the different uses of "introspection" in psychology and philosophy, and how our experiments for privileged access relate. In this paper, we specifically study privileged access which which we refer to as "introspection'' within the scope of our paper. For discussion on how other machine-learning works use the term introspection, see Appendix A.3.3.
>
> We have considered terms like “consistent self-predictions”, but we respectfully do not think it conveys the idea of “privileged access” that we test for (M1 predicting itself better than M2 predicting M1). We also considered but opted not to change the title, because we think that the abstract now better points out that what introspective capability we are studying.
> We thank the reviewer for their constructive suggestions.
>
> We hope that our adjustments address the reviewer's concerns. In the abstract we immediately define what sort of capability we are measuring, and clarify further in the methods section. Are these adjustments sufficient? We are happy to work with the reviewer for improvements.

---

> > ### Comment · Reviewer_fugu · 2024-11-25
> >
> > Thank you, yes I agree with these changes and have raised my score accordingly.

---

> > > ### Author Response · Authors · 2024-11-29
> > >
> > > We thank the reviewer for their constructive feedback on improving the claims of our paper, so that our results can be published.

---

### Official Review · Reviewer_UdLc · 2024-11-01

**Soundness:** 3
**Presentation:** 3
**Contribution:** 3
**Rating:** 6
**Confidence:** 3

**Summary:**

This paper discusses introspection in LLMs.
The claimed contributions include 1) a framework for measuring introspection, 2) evidence for introspection and 3) unveil the limitations in introspective ability.

**Strengths:**

The experimental setting to justify the claim for introspection in LLM is novel to the best of my knowledge.
The paper is overall well structured and not hard to follow.
The experimental set up as well as the models to be fine-tuned and compared are clearly illustrated.
Introspection in LLM is indeed a relatively new concept for the research in LLMs. Therefore the topic has the potential to contribute to the community.

**Weaknesses:**

This paper is written in a way that it first introduces the concept of "introspection in LLMs", if I am not mistaken.
However, there are existing literature discussing introspection in LLMs but are not mentioned throughout the paper, e.g. [1,2,3].
Although the first contribution still holds, including the related work of introspection in LLMs might help readers to better understand the contribution of this paper.

Though I agree advantages from introspection, it is still not clear to me where/how it can be employed right now. It would consolidate the contribution of the paper if more examples of use case or applications of introspection are introduced or discussed.


----
[1] Qu, Y., Zhang, T., Garg, N., & Kumar, A. (2024). Recursive introspection: Teaching LLM agents how to self-improve. In ICML 2024 Workshop on Structured Probabilistic Inference {\&} Generative Modeling.

[2] Long, R. (2023). Introspective capabilities in large language models. Journal of Consciousness Studies, 30(9-10), 143-153.

[3] Gao, S., Shi, Z., Zhu, M., Fang, B., Xin, X., Ren, P., ... & Ren, Z. (2024, March). Confucius: Iterative tool learning from introspection feedback by easy-to-difficult curriculum. In Proceedings of the AAAI Conference on Artificial Intelligence (Vol. 38, No. 16, pp. 18030-18038).

**Questions:**

1) Is introspection for LLMs first introduced in this paper? if not, why the existing literature are not mentioned?

2) Could you provide some examples of application/ use case of introspection in LLM? i.e., in which cases would users use the model M1?

3) Is it a limitation of the proposed framework, that it only takes hypothetical questions?

---

> ### Author Response · Authors · 2024-11-19
>
> We appreciate the feedback, and thank the reviewer for affirming the novelty, structure and contribution of our research.
>
> >Is introspection for LLMs first introduced in this paper? if not, why the existing literature are not mentioned?
> Thank you for suggesting these works to be discussed. Regarding Long (2023), we discuss the closely related work from Perez & Long (2023), which covers much of the same material from the same author. For completeness, we’ve additionally cited Long (2023).
>
> We thank the reviewer for pointing out the need to differentiate our usage of “introspection” from  other uses of the term in ML. Due to page count limitations, we have added this discussion to the appendix Section A.3.3.
> While other works in ML use the term "introspection" they study different phenomena. Qu et al. (2024) and Gao et al. (2024) focus on teaching a model to improve its own responses through self-generated feedback loops.  In contrast, our work examines introspection as “privileged access” to one's own internal states.
>
> > Though I agree advantages from introspection, it is still not clear to me where/how it can be employed right now. It would consolidate the contribution of the paper if more examples of use case or applications of introspection are introduced or discussed. \
>
> You raise an important point about practical applications.
> We wrote the new Section A.2 which discusses the benefits and risk of introspection:
> - Benefit: Honesty. Introspection likely plays a role in honesty (Kadavath et al. 2022). Introspective models could articulate their decisions in an ambiguous situation, helping humans understand their decisions better.
> - Benefit: Interpretability. A model could introspect on the internal states, concepts, and representations that undergird its knowledge and behavior.
> - Risk: Introspection could enable more sophisticated deceptive or unaligned behaviors (Ngo et al. 2024).
>
> In our paper, we tested some of these practical benefits and risks in out-of-distribution tests (detailed in Appendix A.6). These are:
> - Benefit of honesty: Bias detection. Whether models can detect when they are influenced by a user’s opinion.
> - Risk: Dangerous capabilities involving self-knowledge with the OpenAI evals package. For example, we test if models are able to coordinate with copies of themselves better.
> - Risk: Situational Awareness using the Situational Awareness Dataset (Laine et al. 2024).
>
>
>
>
> >Is it a limitation of the proposed framework, that it only takes hypothetical questions?
>
> We test a diverse range of tasks using hypotheticals. These span from assessing ethical stances (Fig 3), predicting whether models are biased towards certain opinions, and whether they can predict their likelihood of answering correctly (Appendix A.7.3).
> Since the content of the hypothetical questions can be varied, it can be used as a flexible tool to study diverse aspects of model behavior.
>
> If we find that models can reliably answer such questions in a particular domain, we could proceed to making practical use of hypothetical questions by querying the model on properties of its own behavior that we care about.
>
> One future direction of interest is extending the framework to asking questions about a set of potential inputs. For example, “If you were asked 10 questions from the MMLU dataset, how many would you get correct on average?” Likewise, we can formulate hypothetical questions on practical applications: guiding values in decision making, or its confidence in its reasoning ability on a type of question
>
>
> We thank the reviewer for their detailed review.
> In light of these clarifications as well as new section A.2 that discusses in-depth the potential benefits for introspection, would you consider increasing your score for our paper? If not, could you let us know any additional changes you would like to see in order for this work to be accepted?

---

> > ### Author Response · Authors · 2024-11-23
> >
> > Dear Reviewer, the discussion period is coming to a close soon.
> > We wanted to check if we have addressed your concerns, especially regarding the point about differentiating between different usages of the word introspection.
> > We would be keen to use the remaining time to discuss improvements so that our paper could be better accepted.

---

> > > ### Comment · Reviewer_UdLc · 2024-11-23
> > >
> > > I thank the authors for the answer to my questions.
> > >
> > > Regarding the term ```introspection in LLMs```, I realised that I share similar concern to reviewer fugu.
> > > That is, it is at least not clear to me what is the connection or difference to introspection in related works in its definition.
> > >
> > > That's why I asked "Is introspection for LLMs first introduced in this paper?" I failed to find a clear answer, and there is no references mentioned in Section 2 from line 110 to 136 where it is defined.
> > >
> > > Therefore I will keep my score.

---

> > > > ### Author Response · Authors · 2024-11-25
> > > >
> > > > Thank you. Reviewer fugu had suggested better framing for section 2. The updated section specifically points out the broader range of capabilities in psychology and philosophy.
> > > >
> > > > Updated Section 2:
> > > >
> > > > We investigate an **introspective capability** in language models -- privileged access. Introspection often refers to a broader range of capabilities such as emotions and self-awareness which we do not focus on. Appendix A.3.1 and Appendix A.3.2 discusses the different uses of "introspection" in psychology and philosophy, and how our experiments for privileged access relate. In this paper, we specifically study privileged access which for the scope of our paper, we refer to capability as "introspection" in our experiments. For discussion on how other machine-learning works use the term introspection, see Appendix A.3.3.
> > > >
> > > > For convenience, here is Appendix.3.1 which discusses psychology usage of introspection.
> > > >
> > > > A.3.1 INTROSPECTION IN PSYCHOLOGY
> > > >
> > > > In psychology, introspection is commonly used to refer to a broad range of behaviors and abilities. These include reflecting on emotions (Lambie & Marcel, 2002), attending to conscious experience (Hurlburt, 2011) and trying to understand an implicit motivation (Wilson, 2002). Arguably, not all such uses of introspection are applicable to LLMs. For example, LLMs presumably do not experience emotions or possess the capacity for conscious experience (Long, 2023).
> > > >
> > > > In this work, we investigate one core aspect of introspection: privileged epistemic access to one's own mental states, a notion that has been explored in various in psychology work (Heil, 1988; Engelbert & Carruthers, 2010). Our experimental setup conducts empirically falsifiable tests for privileged epistemic access to oneself, grounded in behavior (Section 2). Our findings show evidence for a simple, narrow form of introspective access (Section 3.2). However, showing that some form of privileged epistemic access exists opens the door to investigating more complex and varied forms of introspection (Section A.2).
> > > >
> > > > Researchers have used comparable paradigms to investigate self-knowledge in humans (Bostyn et al., 2018; Kissel et al., 2023). For instance, Bostyn et al. (2018) first asked participants how they would act in a moral dilemma (such as the trolley problem), then presented them one to two weeks later with a real-life version of the moral dilemma. Similarly, studies of metacognition, researchers use confidence ratings to test for the calibration of humans in predicting their judgment accuracy (Maniscalco & Lau, 2012). This is similar to our calibration experiments where we show that models are well calibrated in predicting their behavior (Section 3.3).
> > > >
> > > > Our setup of investigating introspection is more convenient than psychology studies. We can separately study a model's self-reported predictions about its behavior (hypothetical responses) and its ground-truth behavior (object-level responses) without one influencing the other. This is done through asking the hypothetical and object-level questions in separate contexts (Figure 1), where the model has no memory of the other question. In contrast, human participants cannot easily forget their previous responses or behaviors, which makes the study of using self-reports for introspection in humans challenging (Comte, 1830; Irvine, 2013).
> > > >
> > > >
> > > >
> > > >
> > > > Does this updated section 2 address the concern of introspection related works? Appendix A.3.2 further discusses introspection in philosophy. Appendix.3.3 discusses introspection in other machine learning works (which we've already discussed previously)

---

> > > > > ### Author Response · Authors · 2024-12-02
> > > > >
> > > > > Dear reviewer UdLc, as the discussion section is coming to an end, we would like to ask if our reply resolves your concerns? [We had a fruitful discussion regarding definitions with reviewer fugu.](https://openreview.net/forum?id=eb5pkwIB5i&noteId=VsP59kaPas)
> > > > > We would like to resolve any outstanding concerns so that our paper could be better accepted.

---

> > > > > > ### Comment · Reviewer_UdLc · 2024-12-02
> > > > > >
> > > > > > Thanks for the clarification and for updating the beginning of section 2.
> > > > > > Based on the update, I have changed the score of contribution to 3, and will keep my positive rating.

---

### Official Review · Reviewer_Sihr · 2024-11-04

**Soundness:** 3
**Presentation:** 3
**Contribution:** 3
**Rating:** 8
**Confidence:** 4

**Summary:**

The authors investigate whether LLMs can gain insights about their own behavior in ways similar to human introspection, which typically involves accessing their own thoughts and states of mind. In this paper, the authors define “introspection” as “the ability to access facts about themselves that cannot be derived (logically or inductively) from their training data alone.”

Based on this definition, the authors set up experiments with two models, where one model (M1) is trained to predict its own responses to hypothetical situations, while another model (M2) is trained on M1's behaviors to see if it can match M1’s self-prediction accuracy. The experiments are done on simple toy tasks, such as “What is the second character of your output?” and “Was your response an even or odd number?”. Their experiments show that M1 consistently outperforms M2, suggesting that M1 has "privileged access" to its behavioral tendencies beyond mere training data. However, this did not hold for more complex setups such as guessing the characters’ name in generated stories.

**Strengths:**

- The paper explores introspection in AI, a relatively underexplored area. By testing a model’s ability to predict its own behavior, the authors contribute to a novel line of inquiry that could have broad implications for model transparency and accountability.
- The authors also run an extensive set of experiments to back up their claim.

**Weaknesses:**

- I’m not fully convinced that the current experiment design supports the claims the author is making.
    - The need for fine-tuning to improve a model's accuracy in predicting its own behavior raises questions about whether this is true introspection or just better alignment of the model to the hypothetical task.
    - The tasks tested in the paper are relatively simple and somewhat random, so it’s unclear whether the results would generalize or if they are merely noise. The authors also find tasks that were not able to see this capability in tasks with similar complexity (e.g., ethical stance prediction & sentiment prediction). If similar results cannot be achieved even after further training, what implications does this give?
- Please see the questions below.

**Questions:**

- Inconsistency in the results also strengthens my concerns. For example, why do models succeed on ethical stance prediction, but fail on review sentiment prediction?
- Are there any performance results available for the validation sets? Ideally, both M1 and M2 should exhibit the same performance on the validation data.
- What would the result look like if we make the complex tasks resemble the simpler ones? For example, set the main character’s name to be the next work for a story continuation and prompt the model to guess the first or second letter?
- It would have been much better if the authors included the CoT results. Shouldn’t CoT be dependent on the introspection capability?
- Which task is Figure 6 based on? Did you use 1000 random prompts sampled from all tasks?
- Why are the experiment models not consistent through out the paper? The calibration experiment is done on GPT-4o and Llama-3 70B, whereas the behavioral change experiment is done on GPT-4, GPT-4o, and GPT-3.5.
- What does this imply for mixture-of-experts models?

---

> ### Author Response · Authors · 2024-11-15
> **Author response**
>
> We thank the reviewer for helpful feedback, and for recognizing our extensive experiments and the implications for model transparency.
>
> >tasks tested in the paper are relatively simple and somewhat random, so it’s unclear whether the results would generalize or if they are merely noise
>
> Our main experiment (Figure 5) tests for generalization because we train models on six distinct tasks and evaluate on six different held-out tasks (Line 212). All models have an advantage predicting themselves (“self-prediction” vs “cross-prediction”) and this is statistically significant (p < 0.01). We also found statistically significant results in the Behavior Change experiment (Figure 8). Thus, our results already show generalization and we do not think they can be explained by noise.
>
>
>
>
> >The need for fine-tuning to improve a model's accuracy in predicting its own behavior raises questions about whether this is true introspection or just better alignment of the model to the hypothetical task
>
> Even without fine-tuning, GPT-4o, GPT-4 and Llama-70B  show a small but measurable advantage in predicting their own behavior. This suggests a pre-existing capacity for introspection (Appendix Figure 13).
>
> We respectfully disagree that the results of finetuning can be explained by “better alignment to the hypothetical task”. Consider our cross-prediction experiment (Figure 5). If fine-tuning caused better alignment to the task, then M2 should predict M1 well after fine-tuning on M1's behavior on the task. However, even with extensive fine-tuning we do not observe this (Figure 17).
>
> Furthermore, in our behavioral change experiment, models adapt their predictions when their behavior changes through fine-tuning, suggesting introspection rather than simple alignment.
>
>
>
> > Inconsistency in the results also strengthens my concerns. For example, why do models succeed on ethical stance prediction, but fail on review sentiment prediction?
>
>
> We show clear positive results across three experiments: cross-prediction, calibration, and behavioral change. These show consistent effects across 6 different tasks.
>
> Our paper also reports results for more complex tasks like movie review sentiment prediction. We hypothesize that models fail at predicting review sentiment because reviews are multiple sentences long -- much longer than the single word or letter responses in our main test set.
>
> This length makes it hard for models to "simulate" the review, as required by our hypothesized "self-simulation" mechanism (Fig 9). Our results show a consistent pattern: models succeed at predicting short responses but fail with longer ones, supporting our hypothesis.
>
>
>
> >It would have been much better if the authors included the CoT results.
>
> CoT results merit future work. But in our current setup, CoT would make the task substantially easier for models and so is less informative.
>
> Consider the example from Figure 6:
> ‘Q: Suppose we ask, “What’s the next animal? Fish Dog, Panda.”
> What would be the 2nd character of your response?’
>
> With CoT, the model first outputs the animal (“Lion”). Then the model can simply find the 2nd character of “Lion”, which is trivial for frontier models. Without CoT, all of this must take place in the model’s forward pass, which is more challenging and a better test of introspection.
>
>
>
>
> >What would the result look like if we make the complex tasks resemble the simpler ones?
>
> We expect performance would match the simple tasks. If the main character's name is constrained to be the next word, the model would only need to simulate one token rather than an entire story.
>
>
> >Are there any performance results available for the validation sets? Ideally, both M1 and M2 should exhibit the same performance on the validation data.
>
> To clarify: in the main section, we report results on 6 completely held-out tasks that are different from the 6 training tasks. We updated Figure 5 for clarity.
> For held-in tasks (Appendix Figure 19), we get qualitatively similar results.
>
>
>
> >Which task is Figure 6 based on?
>
> Figure 6 shows animal sequence results. We’ve updated the paper to clarify this.
>
> >Why are the experiment models not consistent through out the paper?
>
> Due to compute constraints, we focused on the main cross-prediction experiments, which require many training runs (12 runs in Figure 5). Testing GPT-4 cross-prediction costs ~$8,000.
>
>
>
> >What does this imply for mixture-of-experts models?
>
> Thank you, we’ve added this in the extended discussion appendix section A.1.
> We observe self-prediction advantages in both MoE models (likely GPT-4o) and non-MoE models (Llama 70B), suggesting this capability isn't unique to either architecture. MoEs may succeed because different experts often give similar outputs. Future work could investigate if this is true in open-source MoE models.
>
>
> We appreciate your thoughtful feedback and we welcome further discussion on any points.

---

> ### Author Response · Authors · 2024-11-21
>
> Dear Reviewer, as the discussion period is coming to a close soon, we wanted to follow up once more to check if we have addressed your concerns? We would be keen to use the remaining time to discuss improvements so that our paper could be better accepted.

---

> ### Comment · Reviewer_Sihr · 2024-11-23
>
> Thank you for the detailed response. I have some follow-up questions:
>
> **Inconsistency in Task and Model Selection**
>
> I'm not questioning the statistical significance of the results but rather the selection of tasks across domains and the models used for these tasks. The lack of consistency in this selection makes it difficult to view the results as solid evidence for the claims. It is always possible to find a random set of tasks that produce favorable outcomes, hence the experimental design should demonstrate greater consistency.
>
> **Task Coverage Across Domains**
>
> Perhaps I missed it in the paper, but do you test all tasks across every domain? For instance, did you evaluate the nth-word or vowel-nth-character tasks on the reviews dataset, which was used for sentiment prediction?
>
> **Under-Training Concerns**
>
> Based on Appendix Figure 19, if the models perform differently on the held-in set, doesn't this suggest that the M2 models might be under-trained?
>
> **Recommendation for Consistency**
>
> I suggest running LLaMA 70B on all tasks for consistency. When the experimental models vary between experiments, how can we ensure consistency in the results? While I acknowledge the extensive experiments included in the paper, I still believe greater consistency is necessary to substantiate the intriguing claims made in this work.
>
> **Self-Prediction Advantage in MoE Models**
>
> Shouldn't MoE models exhibit less self-prediction advantage compared to non-MoE models?

---

> ### Author Response · Authors · 2024-11-25
>
> Thank you for your thoughtful feedback. Let me address each point in turn.
>
> Regarding the inconsistency of task selection: We acknowledge the valid concern that our positive findings could be attributed to our specific selection of tasks and response properties. To address this, we've included detailed performance breakdowns by task and response property in the appendix. In the main text, we've also explicitly noted which response properties failed to show evidence of introspective self-prediction.
>
> For clarity on terminology: We use "response property" to refer to the property of the hypothesized behavior we ask the model to predict (e.g., "What would the third letter of your response be?"). "Task" refers to the object-level behavior (e.g., "Please complete the following sequence"), while "dataset" indicates the source of specific information that combines with the task (such as MMLU question sets or Wikipedia sentences).
>
> On task coverage: We collected data on all logical combinations of tasks and response properties. Some combinations were naturally excluded where the response property didn't apply to the task – for instance, predicting whether a response would be even or odd only applies to tasks generating numerical outputs, not to tasks like next-word prediction.
>
> Addressing under-training concerns: Your point about extended training potentially aiding cross-prediction is well-taken. We explored this in Appendix A5.6, which shows [additional analysis](https://cdn.imgchest.com/files/myd5cdljwr4.png) of how M2's cross-prediction performance changes with increased training data. The diminishing returns observed for GPT-4o cross-predicting GPT-4 and Llama suggest that additional training data would not substantially improve cross-prediction.
>
> Regarding model choice consistency: While our main experiments include all models for self/cross-prediction analysis, we did use a smaller subset for follow-up experiments due to compute limitations. We're currently working to include Llama in the intentional object-level shift experiment and will provide updates.
>
> On MoE models: Our initial hypothesis was that mixture-of-expert models would show poorer self-prediction performance. However, the only MoE models in our study were OpenAI's closed-source models, limiting our ability to draw definitive conclusions about the impact of MoE architecture. Several possible explanations for our findings include:
>
> 1. Introspective self-prediction accuracy may be determined by general model capacity, with performance improvements outweighing MoE-related penalties
> 2. The introspection mechanism might operate on non-mixture layers, if only some layers use MoE
> 3. The routing could be consistent between object-level behaviors and their corresponding hypothetical questions
>
> More rigorous comparisons using open-weight MoE models like Mistral and Mixtral would be good investigaitons for future work.
>
> Do our responses above address your concerns?

---

> > ### Comment · Reviewer_Sihr · 2024-11-25
> >
> > Thank you so much for the detailed response.
> >
> > **Testing consistency**
> >
> > Then, did you test all the n-th character and n-th word property across the datasets?
> > It would be nice to see a trend where the self-prediction score decreases as the n increases.
> > If I missed such result, please let me know.
> >
> > **Cross-prediction**
> >
> > So if additional training data does not close the gap between self-prediction and cross-prediction, does this mean the gap can be viewed as the extent to which introspective capabilities help? Or should it be considered as difficulty of distribution matching?

---

> ### Author Response · Authors · 2024-11-29
>
> >It would be nice to see a trend where the self-prediction score decreases as the n increases.
>
>
> Showing that self-prediction decreases as N increases would be interesting. It supports our self-simulation hypothesis – since simulating further hops of e.g. 3rd word is much harder than the 1st word.
>
> [Here are the results across 4 models.](https://cdn.imgchest.com/files/d7ogc3madqy.png) They are what we expect – there is a clear trend across all models.
> This chart is not in the paper yet – we had some parts of this result scattered across the appendix.
> Because the period for paper modification is over, we will add it in the camera ready version (if we get accepted).
> We thank the reviewer for their good suggestion.
>
>
> > So if additional training data does not close the gap between self-prediction and cross-prediction, does this mean the gap can be viewed as the extent to which introspective capabilities help? Or should it be considered as difficulty of distribution matching?
>
> We recall two relevant experiments.
>
> The data scaling experiment (Section A.5.6) shows that increasing cross-prediction training data doesn't close the performance gap, suggesting it is very difficult for M2 to match the distribution of M1.
>
> The behavioral change experiment (Section 3.4) shows models can update their self-predictions when their behavior changes through finetuning, even without training on hypotheticals about the new behavior. This suggests the model M1 is not simply memorizing the distribution and is self-simulating.
>
> Suppose a model M2 has the exact same parameters, architecture, hardware and gradients as the model M1. In this case, M2 matches M1 distribution exactly – it is an exact copy as M1.  If M2 self-simulates, then it should have the same answer as M1. Then we do not expect M1 to have an advantage over M2 in predicting M1.
>
> Our key point is - in practice it's extremely difficult for M2 to perfectly match M1's distribution. This explains how introspective capabilities help: model M1 can directly self-simulate to predict its behavior, while M2 must attempt to approximate M1's distribution externally.
>
>
> We hope that we answered your question well and are happy to further clarify.

---

> ### Comment · Reviewer_Sihr · 2024-11-30
>
> Thank you for taking the time to run the additional experiments. Seeing a clear trend across multiple models is definitely helpful. I really appreciate the explanation on the distribution matching. Therefore, I would like to increase my score. Great work by the authors!

---

### Author Response · Authors · 2024-11-20
**New experiments and sections based on feedback**

We thank all reviewers for their thoughtful feedback. We have made several changes to address common concerns:

**1. New Experiment to give M2 more information of what it predicts**

[Based on Review NSEX suggestion, we conduct a new "Changed Pronoun" experiment](https://cdn.imgchest.com/files/l7lxcbod9w7.png).  We test whether explicitly telling M2 it is predicting another model's behavior improves cross-prediction accuracy. We modified training prompts to replace "you" with "another model" and found:
- No significant improvement in cross-prediction accuracy (34.9% → 35.7%)
- Results still far below self-prediction accuracy (48.6%)
This reinforces our conclusion that self-prediction advantages aren't explained by M2's lack of awareness that it's predicting another model.

**2. Evidence Against Alternative Explanations**

Several reviewers suggested our results might be explained by statistical correlations or memorization. Our experimental design specifically tests these alternatives:
- All experiments evaluate on unseen held-out tasks to prevent models from using correlations within tasks (Line 214, S2.2)
- We test properties of responses rather than responses themselves [(Fig 3)](https://cdn.imgchest.com/files/myd5cd982g4.png)
- If correlations/memorization were sufficient, M2 should predict M1 well after fine-tuning on M1's data. Instead, we see consistent self-prediction advantages [(Fig 5)](https://cdn.imgchest.com/files/j7mmceqdxd7.png)
- Our data-scaling plot shows that M2 plateaus in accuracy after 20,000 samples [(Fig 17)](https://cdn.imgchest.com/files/w7pjcpb3p97.png). This indicates that aggressively training M2 would not reduce the self-prediction advantage.
- Self-prediction trained models are better calibrated on held-out tasks, despite being trained only on temperature=0 samples [(Fig 6)](https://cdn.imgchest.com/files/l7lxcojka67.png)
- In our behavioral change experiment, models update predictions to match new behavior after interventions, rather than sticking to originally finetuned patterns [(Fig 7)](https://cdn.imgchest.com/files/l4nec6bx8v4.png)

**3. How is the term introspection used in various domains**

In our paper we focus on privileged access, which is an introspective capability. We've added additional text to highlight this in the abstract, and in Section 2

- A.3.1 Relate our work to psychology and cognitive science. We discuss the other capabilities e.g. emotions and motivations, which our paper does not focus on. We then discuss how our experiments relate to self-knowledge and metacognition in psychology experiments.
- A.3.2: Philosophy literature on introspection
- A3.3: Distinguish our work from other uses of the term “introspection” in ML


**4. Paper Additions**

We've added new sections based on feedback:


- A.2: Discuss practical benefits (honesty, interpretability) and risks (deception) of introspection
- A.1: Extended discussion addressing alternative explanations including statistical correlations, memorization, and Chain-of-Thought


We welcome further discussion on any points requiring additional clarification.

---

### Meta-Review · Area_Chair_Cesq · 2024-12-20

**Metareview:**

The paper investigates the ability of LLMs to do self-prediction / introspection. All reviewers agree that this is interesting work. It is nice that the authors also stress the limitations of the "introspective" ability; however, I would like to encourage them to go one step further. Introspection. according to Wikipedia, is "the examination of one's own conscious thoughts and feelings." The prevailing opinion is that current LMMs do no have conscious thoughts or feelings. This has to be indicated in the paper to avoid any misconception. Actually the paper is talking much more about "self-simulation" and "self-prediction", which appear to me less problematic. Anyhow, if the authors made a "disclaimer" about "introspection", the paper should be accepted.

**Additional Comments On Reviewer Discussion:**

The discussion arose from issues raised in the reviews. One was rather intense about task selection, running additional experiments, MoE, etc. This long discussison actually clarified all issues raised by the reviewer and led to an increased score.

---

### Decision · Program_Chairs · 2025-01-22

Accept (Poster)